# Comparing infrared spectroscopic methods for the characterization of *Plasmodium falciparum*-infected human erythrocytes

Agnieszka M. Banas [1✉], Krzysztof Banas [1], Trang T. T. Chu [2], Renugah Naidu[2], Paul Edward Hutchinson [3], Rupesh Agrawal [4], Michael K. F. Lo [5], Mustafa Kansiz[5], Anirban Roy[6], Rajesh Chandramohanadas [2,7✉] & Mark B. H. Breese[1]

Malaria, caused by parasites of the species *Plasmodium*, is among the major life-threatening diseases to afflict humanity. The infectious cycle of *Plasmodium* is very complex involving distinct life stages and transitions characterized by cellular and molecular alterations. Therefore, novel single-cell technologies are warranted to extract details pertinent to *Plasmodium*-host cell interactions and underpinning biological transformations. Herein, we tested two emerging spectroscopic approaches: (a) Optical Photothermal Infrared spectroscopy and (b) Atomic Force Microscopy combined with infrared spectroscopy in contrast to (c) Fourier Transform InfraRed microspectroscopy, to investigate *Plasmodium*-infected erythrocytes. Chemical spatial distributions of selected bands and spectra captured using the three modalities for major macromolecules together with advantages and limitations of each method is presented here. These results indicate that O-PTIR and AFM-IR techniques can be explored for extracting sub-micron resolution molecular signatures within heterogeneous and dynamic samples such as *Plasmodium*-infected human RBCs.

[1] Singapore Synchrotron Light Source, 5 Research Link, Singapore, Singapore. [2] Pillar of Engineering Product Development, Singapore University of Technology and Design, Singapore, Singapore. [3] Flow Cytometry Lab, Life Sciences Institute, National University of Singapore, Singapore, Singapore. [4] National Healthcare Group Eye Institute, Tan Tock Seng Hospital, Singapore, Singapore. [5] Photothermal Spectroscopy Corp, Santa Barbara, CA, USA. [6] Bruker Nano Surfaces & Metrology, Goleta, CA, USA. [7] Department of Microbiology & Immunology, Yong Loo Lin. School of Medicine, National University of Singapore, Singapore, Singapore. ✉email: slsba@nus.edu.sg; micrc@nus.edu.sg

Malaria is one of the most common and life-threatening diseases that continues to affect mankind. Every year, an estimated 400,000 deaths[1] occur because of this mosquito-borne infectious disease. Since 2000, the rate of new malaria vaccine trials registered at ClinialTrials.gov has remained steady at about 10 trials each year[2]. So far, only one of them, the pre-erythrocytic vaccine (PEV) product RTS,S/AS01E has proven its safety and effectiveness in reduction of clinical malaria cases in African children. Up to now, the world is still waiting for optimally developed medications to fight the scourge of malaria. One of the hypotheses claims that malaria vaccine development is slowed down by the complexity of the parasite's life cycle and its vast repertoire of polymorphic proteins.

Early diagnosis is still the key to start immediate treatment and through this, to reduce malaria-associated deaths. Currently, methods such as microscopic examination of infected blood smears[3,4], gene amplification techniques[5,6] or serological detection tests[7] are relied upon in clinical diagnosis of malaria. Some of these techniques require sophisticated instruments, expensive reagents or trained personnel, whereas microscopic methods are easy to perform and cost-effective, but are prone to human errors.

Amongst the various species of malaria parasites, *Plasmodium falciparum* is responsible for most malaria-associated deaths. The sporozoite stage of the parasite enters the human body through the bite of an infected mosquito and proliferates asymptomatically into merozoites upon reaching the liver (the pre-erythrocytic hepatic stage). Later, the merozoites invade the red blood cells (RBC) to start a ~48h-long intraerythrocytic developmental cycle (IDC). The very first stage of this process is constrained to the parasitophorous vacuole (PV), a structure allowing a parasite to grow within cell and at the same time being protected from the cell defense mechanisms[8]. The young parasites, also known as the "ring" stages, have low metabolic activity. To facilitate its development, erythrocyte cytoplasm containing human hemoglobin is ingested into an acidic digestive vacuole (DV). Hemoglobin is digested in these compartments; this process leaves insoluble reactive products due to the release of heme, which is neutralized by the formation of β-hematin that biocrystallizes to the chemically inert hemozoin, also known as the malaria pigment[9,10]. DV plays a crucial role in parasite development, it is also a place for amino acid transportation, oxygen radical detoxification and drug accumulation[11,12]. Along with parasite development, a single large DV is formed at the late ring stage and increases its size during development into trophozoites when maximum hemoglobin digestion is observed. In the next IDC stage, known as schizonts, the DV shrinks in size and hemozoin is released into the blood circulation.

Parasites have unique biochemical signatures representing their cellular composition (nucleic acids, lipids, proteins) which can undergo dynamic changes during the course of IDC. Given the relatively simple composition and structural features of terminally differentiated host RBCs, significant chemical differences between infected and uninfected RBCs, spectroscopic methods could be used to analyze the parasite's developmental process.

Label-free techniques such as Raman imaging microscopy has been tested as a potential method to diagnose *Plasmodium* infections of human RBCs on the basis of strong scattering from the hemozoin pigment[13]. However, a rather low signal to noise ratio, high autofluorescence and potential photodamage limits its effectiveness in revealing molecular structures of infected single RBCs.

Other label-free methods such as tomographic phase microscopy (TPM) were also previously explored to study *Plasmodium*-infected RBCs as it allows visualization of intracellular organelles, membranous structures and life stage transitions from 3D refractive index tomograms[14–16].

Another commonly used method for analysis of *Plasmodium*-infected RBCs is FTIR (Fourier Transform InfraRed) microspectroscopy, a method known since the middle of the 20th century and broadly applied in biomedical research. FTIR microspectroscopy probes intrinsic molecular vibrational frequencies of bonds between molecules present in a sample. It is worth emphasizing that IR light used for the experiments is too weak to cause any cell damage. Typical values of the power of IR emitted from a conventional (Globar) source or synchrotron based source are of ~80 μW over the whole spectral range.

Since vibrational frequencies depend on the parameters' characteristic of the molecular structures, they can provide valuable information on cellular biochemical changes through relative quantification of lipids, proteins, carbohydrate etc, as reflected by published literature discussing analysis of cells by means of FTIR spectroscopy[17–19].

However, it is worth emphasizing that there are shortcomings associated with FTIR, which may prevent its use in single cell-based studies. Theoretically, single cells can be analyzed either when apertures are used in the IR microscope or when the IR microscope is equipped with a 2D Focal Plane Array (FPA) detector. Aperture-based systems considerably limit the light throughput, and hence significantly degrade spectral signal-to-noise (sensitivity), as the aperture masks throw away most of the incoming infrared light. More advanced FTIR microscopes with FPA detectors remove the need for apertures, with the FPA pixels themselves defining measured area. A FPA detector is composed of many basic detector units recording a full FTIR spectrum within the interval 4000–900 $cm^{-1}$ from a defined spot (few $μm^2$) on the sample. However, the amount of light absorbed in the IR region entering the single detector unit is small, leading to noisy spectra. Widely-used FPA detectors composed of 128 horizontal pixels and 128 vertical pixels working with 15x objective lens (Bruker 0.4 NA) have an effective pixel size of 2.7 by 2.7 $μm^2$. However, it does not mean that during experiments chemistry from that spot is recorded—it is only a "pixel size." Pixel size must not be confused with spatial resolution, as it is one of the most critical measurement parameters in FTIR microspectroscopy[20]. The spatial resolution is restricted by the so-called diffraction limit and can be calculated using the Rayleigh criterion, which depends on the wavelength. For 15x objective lens mentioned earlier, simple calculations show that for selected wavenumbers much larger values than 2.7 μm have been obtained: 15 μm (at 1000 $cm^{-1}$), 9.2 μm (at 1650 $cm^{-1}$), and 5.4 μm (at 2800 $cm^{-1}$). It means that the response from the system comes from an area much bigger than the pixel size, leading to spatial averaging of biochemical heterogeneities across areas of up to ~15 μm within the single cell, thus severely limiting intra-cellular imaging.

With new and evolving parasitic forms, drug resistance and lack of a commercial vaccine, there is a pressing need for understanding the interactions of the *Plasmodium* parasites with its host cells during the complex multistage life cycle, towards novel therapeutic interventions. Studying changes that the cell undergoes also has the potential to identify key biological and physiological processes that could lead to new and improved antimalarial treatments. If vibrational spectroscopy is being considered as a potential method of examining infected RBCs, according to the authors of this paper, it is necessary to remember that detailed analysis of parasite-host interactions should be performed at the microscale or even the nanoscale. Multiple small DVs which are characteristic of the early and mid-stage rings have a diameter of 150–300 nm[21], at the trophozoite stage, when hemoglobin degradation is noticeable, DV has a diameter of up to 2.2 μm, and later in schizogony, it decreases in size to 0.8–1.2 μm[22,23].

The last decade has seen impressive progress in the development of new techniques based on the interpretation of vibrational frequencies of bonds present in molecules. One of them is AFM-IR (Atomic Force Microscopy - Infrared) spectroscopy. This technique combines the power of infrared absorption spectroscopy for non-destructive, label-free chemical identification with the nanoscale resolution obtained by atomic force microscopy and can be successfully applied to study the composition of cells at the nanoscale[24,25]. AFM-IR has been already used to analyse malaria infected RBCs[26]. Another novel and highly versatile technique, Optical Photothermal Infrared (O-PTIR) spectroscopy, has captured recent attentions for its versatility and offers submicron resolution for IR imaging and spectroscopy[27]. This method is also label-free, requires no special sample preparation steps, operates in an easy-to-use far-field mode (non-contact) and offers submicron resolution for IR imaging and spectroscopy.

In this paper, we determine the relevance of sub-micron resolution towards the analysis of *Plasmodium falciparum*-infected human erythrocytes by means of vibrational spectroscopy. Comparison of two evolving techniques (O-PTIR and AFM-IR) against the more traditional FTIR microspectroscopy based on results obtained for infected RBCs will be presented along with the advantages and limitations of each method. Infected RBCs (iRBCs) were selected as a subject of our analysis due to their heterogeneity arising from the parasite infection. Our work is mainly focused on illustrating all the benefits (if any) of analyzing iRBCs content with sub-micron resolution. For our case study about 20 iRBCs and 20 control (healthy, un-infected) RBCs were taken into account. Conclusions presented in the paper are a cumulative sum of our consistent observations across all cells tested.

To our knowledge, this is the first comprehensive investigation to classify, compare and identify potential methods which could be used for the spectroscopic characterization of single cells in mid-IR region in follow-up studies.

## Results

**Preliminary validation of spectral quality between the three modalities: FTIR microspectroscopy, O-PTIR and AFM-IR spectroscopy.** All three methods discussed in this paper probe intrinsic molecular vibrational frequencies of chemical bonds present in a sample; these absorption bands are displayed as a function of wavenumber [cm$^{-1}$]. However, as was mentioned in Materials and methods, the process of signal generation for each method varies immensely.

In FTIR spectroscopy, IR light passing through a molecular material can be absorbed. Absorbance is given by the Beer-Lambert-Bouguer Law and is directly proportional to the molar concentration of chemical species, molar absorptivity and path length. AFM-IR relies on rapid thermal expansion effect caused by laser pulses, which have to be tuned to the absorption bands present in the sample. The signal measured for this method include not only information about the concentration of chemical species, but also about thermal and mechanical sample properties. The working principle of O-PTIR is based on detecting the localized photothermal effects elicited by the sample upon absorbing of infrared light. For O-PTIR and AFM-IR, as long as the photothermal responses are linear at each sampling location, the resulting spectra could be effectively normalized where the relative contributions of in thicknesses, thermal and mechanical properties are canceled out over the spectroscopic range.

To prove the equivalence of these three methods for IR spectra collection, prior to comparison of the results obtained for individual RBCs, the similarity of resultant spectra collected by means of FTIR microspectroscopy, O-PTIR and AFM-IR was checked. For this purpose, the microtomed section of a cured epoxy adhesive (thickness of ~800 nm) placed on CaF$_2$ was used as the reference.

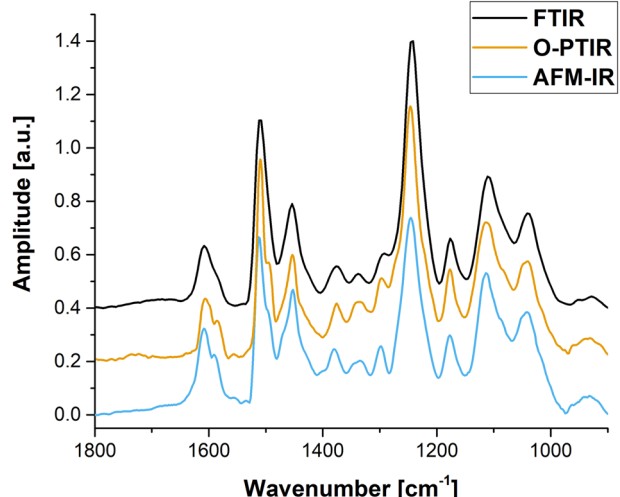

**Fig. 1 Validation of spectral quality between FTIR, O-PTIR and AFM-IR spectroscopy.** FTIR (black), O-PTIR (orange) and AFM-IR (sky blue) spectra collected for a microtomed section of a cured epoxy adhesive (~800 nm) placed on CaF$_2$ window. Raw spectra have been offset for better clarity.

Despite being collected in various modes: reflection (for O-PTIR) or contact (for AFM-IR), spectra presented in Fig. 1, very closely match with the FTIR absorption spectrum measured in transmission mode. It needs to be emphasized that all major absorbance peaks were detected; with their ratios and positions directly comparable.

This finding confirms that since O-PTIR and AFM-IR provide interpretable IR absorption spectra comparable to conventional IR spectroscopy techniques such as FTIR, a discussion about advantages and disadvantages of FTIR microspectroscopy, O-PTIR and AFM-IR has potential interest for scientists dealing with the analysis of single cells.

**Analysis of single, healthy RBC (control) by FTIR microspectroscopy and O-PTIR spectroscopy.** Typical (vibrational) spectrum of a biological sample contains several absorbance bands associated with the main macromolecules, namely amide bands from proteins, membrane lipids (CH$_x$ stretching and deformation), nucleic acids (phosphate stretches), carbohydrates (CO stretching) and phospholipids. The total biochemical composition (and therefore molecular signatures that are extracted by spectroscopic techniques) of *P. falciparum*-infected RBCs (iRBCs) are bound to be different as a result of parasite development (for example, synthesis of *Plasmodium* specific proteins, hemoglobin degradation, peptide and/or amino acid transportation, heme biocrystalization, etc.[12]) and/or arising from metabolism (such as iron deficiency, inflammation[28]). Hence, some spectral differences are expected between infected RBCs and the healthy control cells. These differences can be manifested in shape changes, such as the occurrence of shoulder features in existing bands and/or peak shifts of the main absorbance bands in the collected spectra.

FTIR microspectroscopy was carried out using a FPA detector (128 pixel × 128 pixel); a single pixel element defines a spot size of 2.7 μm$^2$, though the actual achievable spatial resolution is limited by the wavelength dependent Rayleigh criterion. With the microscope transmission mode configuration for the main lipids peaks this is ~5 microns and up to ~15 microns in spatial resolution for the saccharide bands at around 1000 cm$^{-1}$. A full field of view (345 × 345 μm$^2$) is analyzed in one measurement, as its dimension is related to the 15x magnification objective lens attached to the microscope, it cannot be changed.

A single experiment using a FPA detector generates a hyperspectral object with 16,384 spectra. Each spectrum contains information about all absorbance bands present in the sample. By calculating the area under selected bands in the spectra, an image of the spatial distribution for particular infrared absorption is created. This methodology is quite convenient, taking into account that one experiment (taken within 32 min) can generate a complete range of spectra, which can be later used to obtain spectroscopic images for any selected wavenumber (simply by performing mathematical calculations). Many scientists rely on this label-free imaging capability, as it does not require the usage of additional dyes; different chemical components are visualized based on inherent molecular vibrations arising from them. Figure 2a shows a screenshot from the OPUS software (provided with the Bruker instrument); as can be seen, the area selected for experiment (red square) contains not only RBCs, but also substrate material. In order to analyse only one selected RBC, digital zooming is needed.

Amide I ($C=O$ and C–N) at ~1650 $cm^{-1}$ (Fig. 2c) and proteins/lipids ($\nu COO_2$ of fatty acids and amino acid side chains) at ~1391 $cm^{-1}$ (Fig. 2e) distributions are presented only for the zoomed area marked in Fig. 2a by a green square (additionally single cell of interest is marked by a pink square).Values for amide I and proteins/lipids bands were obtained during post-processing (integrals calculation under selected bands) of the collected array of the spectra.

It is known that FTIR microspectroscopy provides an excellent way to visualize the spatial distribution of cellular constituents (lipids, phospholipids, DNA, etc.)[15,16,23]. However, looking at the distribution of amide I and proteins/lipids (presented in Fig. 2c, e), it is clear that the image lacks definition and little to no significant spectroscopic change is observed where the selected cell is present (marked in Fig. 2a by pink square).

Figure 2d, f show results obtained during analysis of the same cell using O-PTIR spectroscopy (Fig. 2b). To better compare FTIR and O-PTIR results, hyperspectral data is discussed. The size of the hyperspectral object was set to 6 by 6 $\mu m^2$ to completely cover the selected cell, 169 spectra were collected with a spacing of 0.5 $\mu m$ in $x$ and $y$ directions. This set of parameters ensures taking full advantage of the wavenumber-independent sub-micron spatial resolution of the O-PTIR technique. In Fig. 2d, f the distribution of amide I and proteins/lipids may reveal detailed structural information within the single cell, suggesting the biochemistry of this cell is not purely homogenous as FTIR results points out. It is known that healthy RBCs are devoid of a nucleus and other organelles typical of eukaryotic cells. And above all, they have a bi-concave shape with thicker exterior and thinner interior parts, so protein and content distribution within a healthy RBC cannot be uniform.

In vibrational spectroscopic measurements, spectra are the most basic yet most informative component for studying biochemical changes among samples. As previously discussed, the process of data collection varies significantly among FTIR and O-PTIR spectroscopy, however their resultant spectra are equivalent and can be used to gain information about the biochemical constituents of the cells.

Figure 2g depicts an example of three consecutive FTIR spectra taken from the points marked in Fig. 2c, e. Spectra 1 are 2 are quite similar, taking into account the intensity and position of the most prominent bands, but some subtle differences are noticeable. For spectrum 3, the reduction in the intensities for the present bands is noticeable, probably recorded from the edge of the cell, as the diameter of the cell is less than 6 $\mu m$ and the nominal dimension of the single FPA detector is 2.7 $\mu m$. Additionally, the spatial resolution of the mid-infrared is wavenumber dependent and is typically in the range of 5 to 20 $\mu m$ per the Rayleigh criterion. Therefore, the said technique

would be providing an averaged biochemical information of the whole cell, lacking resolution to resolve domains smaller than the diffraction-limited spatial resolution.

As a comparison, Fig. 2h shows six spectra taken with the step 0.5 $\mu m$ by means of O-PTIR spectroscopy from the region marked in Fig. 2d, f. As can be seen five of the six spectra reveal the biochemistry of the area smaller than the size of single detector of FPA, but some obvious variations in the positions of amide I and the intensities of other bands related to other cell constituents are observed. Spectrum 6 was recorded 0.5 $\mu m$ away from the cell perimeter showing baseline spectral response from the substrate. This is a remarkable photothermal infrared response providing direct evidence of the submicron spatial resolution capability with O-PTIR spectroscopy.

**Comparison of results collected for infected and control RBCs by means of FTIR microspectroscopy and O-PTIR spectroscopy.** It is difficult to conclude whether lack or presence of visible differences among spectra collected for the same single cell (control RBC) can be treated as a general trend for all analyzed cells. Wider perspective is needed to determine an objective conclusion about the generally observed similarities and differences among a larger set of infected and control sets of RBCs. Therefore, the following analysis was performed: 124 spectra collected from control and 97 spectra from infected RBCs (iRBCs —the trophozoite phase) for FTIR microspectroscopy, for O-PTIR spectroscopy, on the other hand, 64 spectra for control and 164 spectra for iRBCs were taken into account.

Principal component analysis (PCA) was applied to reduce the dimensionality of original variables space (wavenumbers) and transform the source of variability in the data into the first few variables (PCs). This analysis was done to recognize any clustering pattern among studied spectra. All spectra were area-normalized and baseline corrected prior to PCA. Additionally, PCA was performed on centered and scaled data.

Bigger symbols in the scatterplots mark two-dimensional PCA centroids (or barycentres) that are mean scores for each class (Control and Infected) for the first two principal axes.

Shaded colored areas show the confidence ellipses around group mean points (centroids). Each area represents a 95% confidence ellipse for a set of 2D normally distributed data samples and allows visualizing a 2D confidence intervals.

In Fig. 3a, the PCA score plot (PC1 vs PC2) failed to separate the control from the infected RBCs based on the spectra collected by FTIR microspectroscopy. The first and the second component (PC1, PC2) contribute to only 58.5% of data variance in total. Even if the third component (PC3), explaining 8.9% variances is taken into account, it does not allow us to observe any grouping or patterns (Fig. 3b). For the O-PTIR results, a clear division is perceived for the analyzed spectra, not only for PC1 vs PC2 (Fig. 3b) but also for PC1 vs PC3 (Fig. 3d) plots. In Fig. 3b, d, PC1 (explaining 55.9% variances) is mostly responsible for this outcome. It should be stressed that as PCA is one of the unsupervised methods, different symbols and colors for control and infected RBC subsets were chosen by us for better visualization of PCA results.

Comparison of the O-PTIR mean spectra collected for control and infected RBCs with PC1 loading (Fig. 4a) suggests that the following regions can play an important role in observed clustering for O-PTIR measurements: 1664–1500 $cm^{-1}$ (amide I and amide II), 1280–970 $cm^{-1}$ (saccharides, DNA, and phospholipids), 1400–1330 $cm^{-1}$ (lipids, proteins) and 1780–1690 $cm^{-1}$ (lipids). Peaks in loading 1 studied in this work do not resemble the actual spectra because, as was mentioned earlier, PCA was performed on scaled and centered data. It is

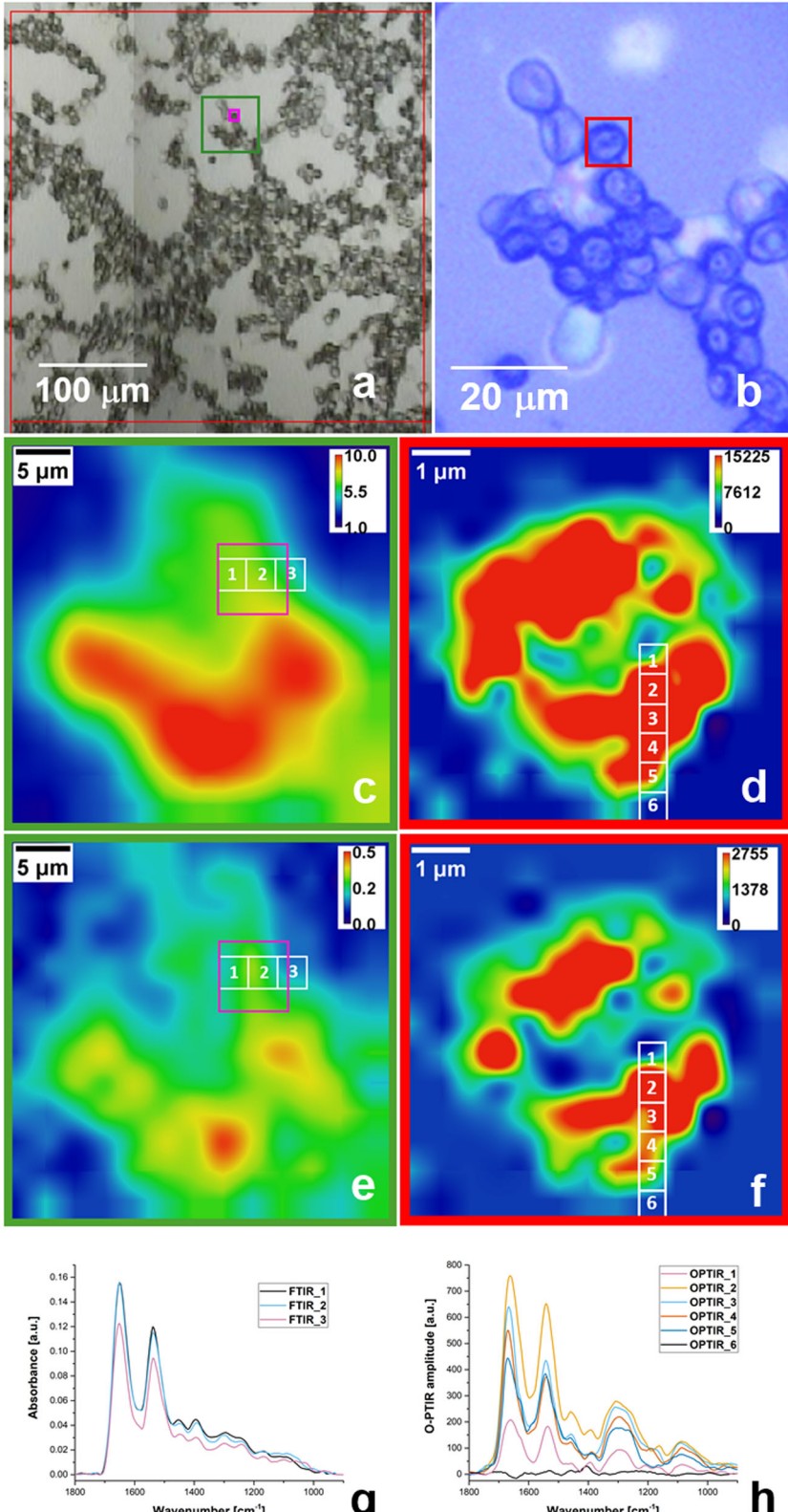

**Fig. 2 Comparison of results for single RBC (control) by FTIR microspectroscopy and O-PTIR spectroscopy.** Screenshots showing the area selected for FTIR microspectroscopy experiment (**a**), red square denotes the area being analyzed during one single FPA measurement (128 pixel × 128 pixel), small pink square marks the position of a selected RBC; and O-PTIR (**b**), red square contains the area chosen for hyperspectral experiment (one single cell). Chemical map reconstructed for amide I and proteins/lipids distributions by means of FTIR microspectroscopy (**c, e**), and O-PTIR spectroscopy (**d, f**). In Fig. 2c and e only zoomed region displaying 13 × 13 of 128 × 128 single elements of FPA including cell of interest is presented (marked by pink square). Comparison of spectra collected by means of FTIR microspectroscopy (**g**) and O-PTIR spectroscopy (**h**). Spectra were recorded from the points marked in **c, e** and **d, f**, respectively. Distance between spectra is equal 2.7 μm for FTIR microspectroscopy and 0.5 μm for O-PTIR.

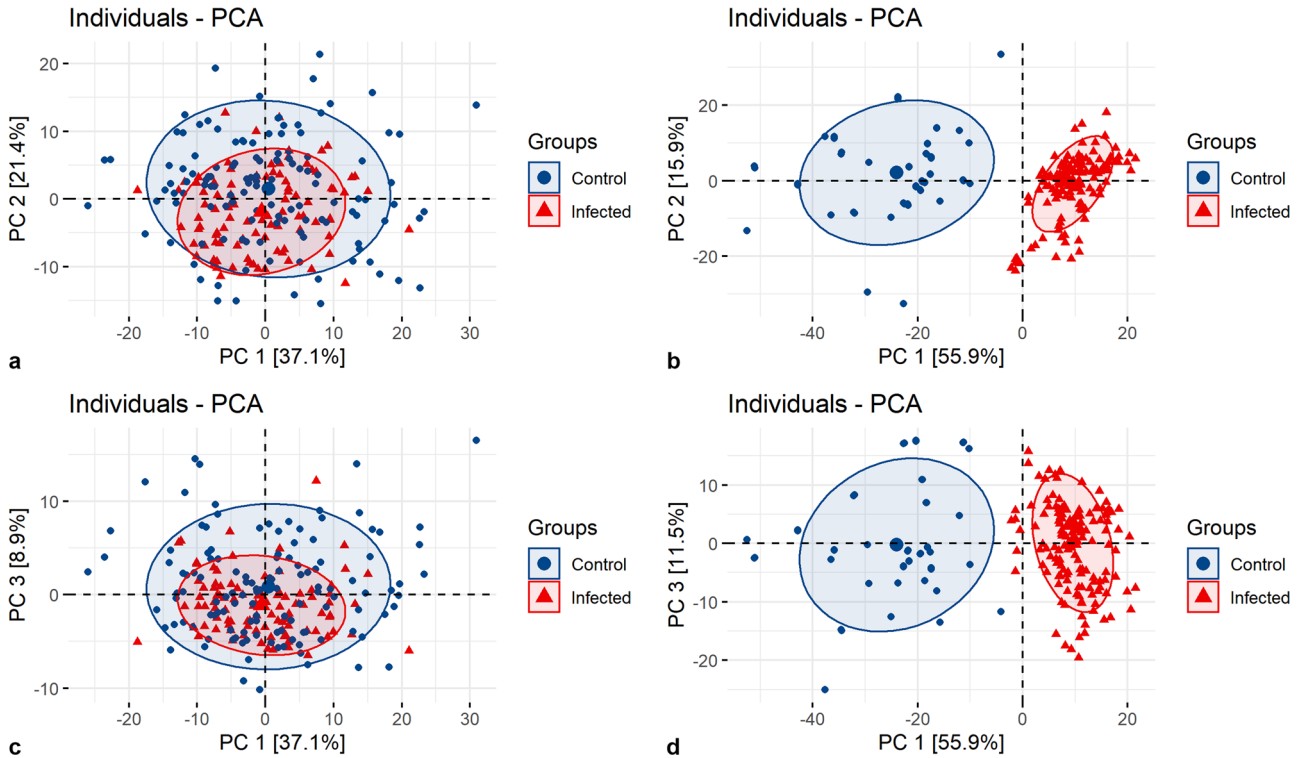

**Fig. 3 PCA on spectra collected for control and infected RBCs by FTIR microspectroscopy and O-PTIR spectroscopy.** PCA score plot along PC1 & PC2 and PC1 & PC3 calculated for spectra collected for control and infected RBCs by means of FTIR microspectroscopy (**a**, **c**) and O-PTIR spectroscopy (**b**, **d**).

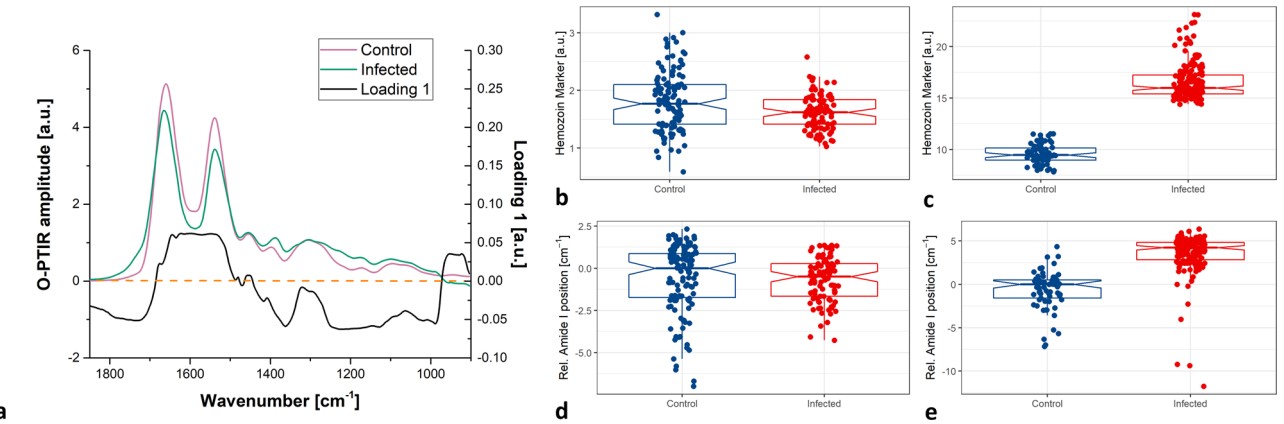

**Fig. 4 Detailed analysis of spectra collected for control and infected RBCs by FTIR microspectroscopy and O-PTIR spectroscopy.** Comparison of the O-PTIR mean spectra collected for control and infected RBCs with PC1 loading (**a**). Boxplots of area calculated under the spectrum line within the region 1720–1708 cm$^{-1}$ for each spectrum collected for control and infected RBCs by means of FTIR microspectroscopy (**b**) and O-PTIR (**c**). Boxplots of the relative positions of the amide I bands in single spectra collected for control and infected RBCs by means of FTIR microspectroscopy (**d**) and O-PTIR (**e**). The lower and upper hinges of the boxplots correspond to the first and third quartiles, middle line to the median value. The whiskers extend from the hinge to the 1.5 * IQR (the distance between the first and third quartiles). The notches extend 1.58 * IQR / sqrt(n), this gives a roughly 95% confidence interval for comparing medians.

critical to standardize the variables prior to PCA because it gives more weightage to those variables that have higher variances. In our case we want to ensure that the highest in the amplitude and in the variance amide I and amide II bands will not dominate the results of the analysis (PCA done on not-centered and not-scaled data shows that 98.7% of variability is included in the first PC).

In our study, all iRBCs chosen for measurements were at the trophozoite stage. Based on literature[12], this is the most biochemically and metabolically active stage where the large amount of hemoglobin is digested by the parasite within DVs and hemozoin known as a malaria pigment is produced. Spectroscopic

studies[28] revealed that hemozoin is similar to its synthetic analog β-hematin possessing bands at ~1720–1708 (strong), 1664 (strong), and 1220–1209 cm$^{-1}$ (weak) related to H-bonded carboxylate group, C=O, and C–O stretching vibration of propionate linkage, respectively.

Searching for direct proof of hemozoin presence in the collected spectroscopic spectra should not be focused only on the most prominent hemozoin band (C=O stretching), as its position at ~1664 cm$^{-1}$ is overlapped with the strong amide I band and its presence is manifested only as a hump in the PC1 loading (Fig. 4a).

Another prominent band (H-bonded carboxylate group at 1720–1708 cm$^{-1}$) can be taken into account in that analysis (as is also suggested by PC1 loading). Figure 4 shows values of the integrals calculated for every analyzed spectrum within the region 1720–1708 cm$^{-1}$ for FTIR microspectroscopy (Fig. 4b) and O-PTIR measurements (Fig. 4c). This value can help in differentiating between spectra collected for control and infected RBCs by means of only the O-PTIR technique (Fig. 4c).

Strong negative band in PC1 loading within 1780–1690 cm$^{-1}$ range, may point not only to a band directly related to hemozoin (1720–1708 cm$^{-1}$), but also at stretching vibrations of C=O in lipids (1750–1735 cm$^{-1}$), which can contribute to its formation.

Analysis of amide I bands can also shed some light on modifications in the secondary structure composition in the iRBCs. This modification can be manifested by changes in band amplitudes and shifts of its peak position. Looking at the individual positions of the amide I bands for both discussed methods, again it can be concluded that a blueshift is mostly visible in spectra collected by O-PTIR spectroscopy (Fig. 4e). The shift can be a result of hemozoin presence (as its strong band is observed at 1664 cm$^{-1}$).

Based on the presented results, we can conclude that sub-micron resolution achieved by O-PTIR spectroscopy made it possible to see the differences among control and infected RBCs.

However, in our work only the cells at the trophozite stage were studied. As mentioned earlier, DVs strongly related to parasite development undergo constant transformation, their size varies from nanoscale (150–300 nm for mid-stage rings) through micron scale (2.2 μm for trophozoite) to submicron/micron scale (0.8–1.2 μm for schizogony stage). So, if nanoscale resolution analysis is needed, AFM-IR spectroscopy could also be taken into account as a method to analyze individual iRBCs.

**Analysis of individual infected RBC by AFM-IR spectroscopy.** To obtain chemical imaging in AFM-IR spectroscopy, the QCL wavenumber was fixed at a value corresponding to the selected major absorbance bands and the AFM-IR tip was scanned over the sample (with the same step in $x$ and $y$ direction set to 25 nm, at 0.5 Hz scan rate). Simultaneously, sample topography (Fig. 5a) and viscoelastic properties were also collected, giving a unique set of topographical, chemical and mechanical sample details. Figure 5 depicts an example of two selected iRBC band distributions: amide I (at ~1660 cm$^{-1}$) and stretching vibrations of C=O in lipids (at 1740 cm$^{-1}$), which can contribute to hemozoin formation. 1740 cm$^{-1}$ was selected as a way of detecting the presence of hemozoin traces; as in O-PTIR experiments, in the average spectrum collected for iRBCs a hump around 1740 cm$^{-1}$ was evident.

In Fig. 5b, c the IR intensity distributions are not homogeneous and to some extent they correlate with the sample topography (collected as the height channel: Fig. 5a). This is understandable, due to the fact that the photothermal IR signal collected for every

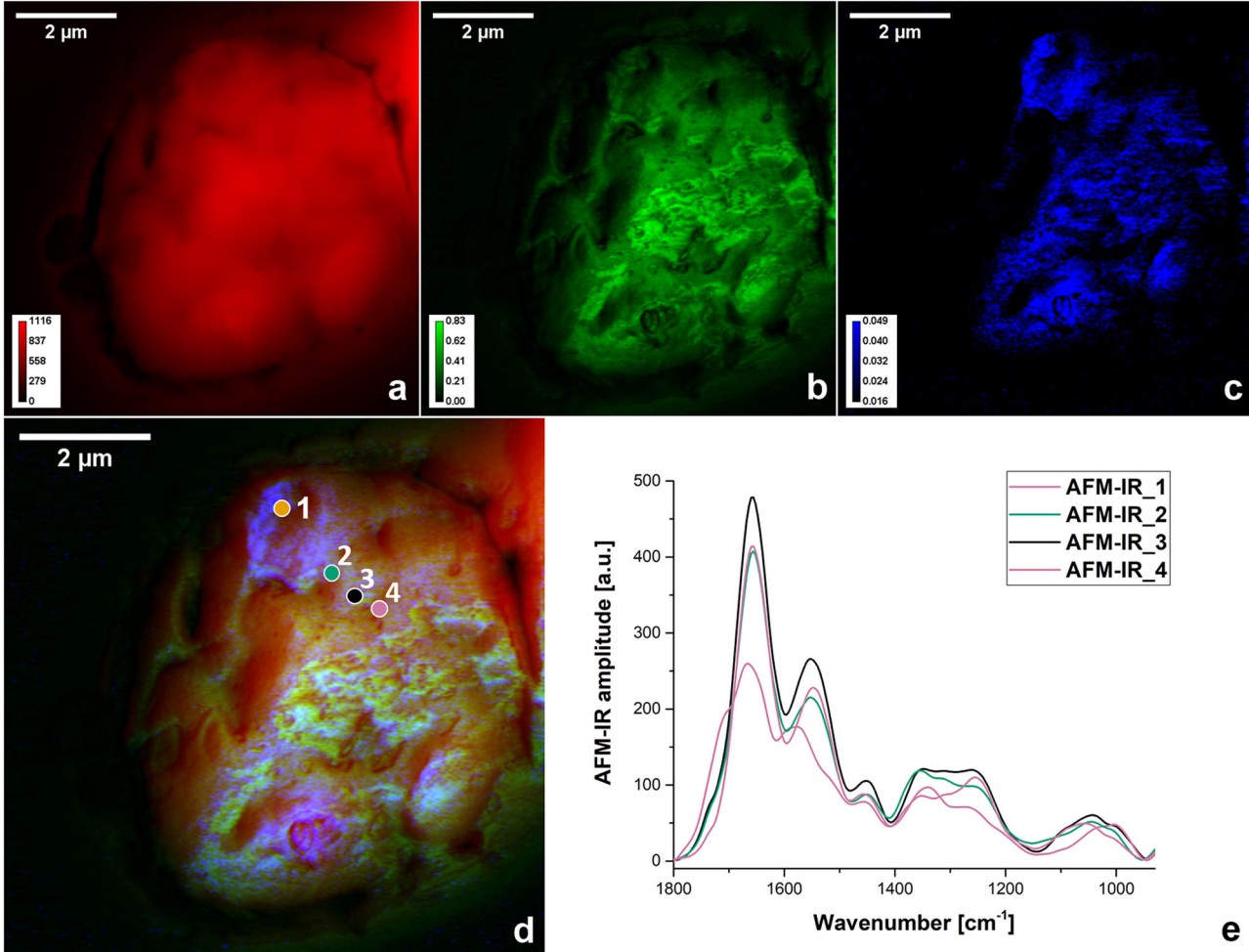

**Fig. 5 Analysis of individual infected RBC by AFM-IR spectroscopy.** Contact mode AFM height image (**a**), distribution of amide I band (**b**) and stretching vibrations of C=O in lipids (**c**) for selected iRBC. RGB-composite image (**d**), showing the AFM height (red) with distribution of amide I band (green) and stretching vibrations of C=O in lipids (blue). AFM-IR spectra (**e**) collected at points 1-4 marked in the RGB- composite image (**d**).

point should show contribution from the area beneath the AFM tip, or more precisely, from the tip-sample interaction volume.

We are aware that presenting only a single distribution of selected bands is not very meaningful especially for a complex sample; to collect AFM-IR chemical image only one wavenumber is chosen, information about changes in its shape, shifts in position are neglected. However, these images can be very useful in finding potentially interesting area to collect spectra that can be used for further analysis. As the spatial resolution for AFM-IR ($\sim$40 nm) means that to cover the area 1 by 1 $\mu m^2$, 625 spectra should be collected, knowing more precisely the region of interest for spectra collection is undoubtedly an advantage.

Figure 5d depicts the way of finding the most promising (from the biological point of view) areas for the single spectra collection. Co-localization of sample topography with selected chemical distributions is presented as a RGB composite image (red channel —topography, green channel—1660 $cm^{-1}$ band, blue channel—1740 $cm^{-1}$ band). It is worth noticing that obtaining such kind of composite image at this spatial resolution is impossible with other available methodologies.

As indicated in Fig. 5c the amplitude of 1740 $cm^{-1}$ band is higher not only in the area surrounding the parasite (compared to a photo taken under a microscope), but also in another part of the cell, which is in accordance with Jaramillo's[29] statement that *Plasmodium* DNA and its hemozoin do not co-localize in iRBC.

AFM-IR spectrum taken at point 1 (marked in the RGB image), where the distribution of 1740 $cm^{-1}$ is dominant, testifies to a presence of hemozoin from noticeable hump at 1708 $cm^{-1}$. Amide I band position is blue-shifted in comparison to amide I position in spectra 2, 3 and 4. Amplitude of amide I is remarkably reduced, what can be related to hemoglobin digestion at point 1 [12]. For spectra 2 and 3 hump at 1740 $cm^{-1}$ is still noticeable, however its amplitude is lower at point 4. Presented spectra (Fig. 5e) point on quite dynamic biochemistry also within the regions assigned to nucleic acids (including the symmetric ($\sim$1080 $cm^{-1}$) and antisymmetric ($\sim$1230 $cm^{-1}$) phosphodiester vibration) and amide III band ($\sim$1286 $cm^{-1}$).

In the AFM-IR method, a sample's thermal expansion due to IR absorption excites the cantilever oscillation of a certain amplitude and frequency. The eigenmodes of the cantilever are related to the contact resonances[24], where the peak frequencies of the cantilever oscillation depend on the elastic properties of the sample: the stiffer the sample, the higher the contact resonance frequency[30].

As the probe scans across the sample surface, the contact resonance of the probe changes, due to the stiffness differences between various sample components, differences in the contact area and force interaction between the tip and the sample. To collect meaningful data, measurements were done with PLL mode (phase-locked loop) enabled. The phase signal from the lockin amplifier was fed to the PLL to track the contact resonance frequency, hence all the nanomechanical information is manifested in the PLL frequency channel, as shown in Fig. 6a. The area corresponding to the iRBC appears to be softer than the surrounding substrate ($CaF_2$ window).

Additionally, quite good correlation (presented in the RGB image—Fig. 6b) between the distribution of the 1740 $cm^{-1}$ band (in red channel) and the PLL frequency (in the green channel) was observed, pointing to the fact that the area with traces of hemozoin are stiffer than the rest of the cell.

## Conclusions
In this paper, we compared three methods: FTIR microspectroscopy, O-PTIR, and AFM-IR spectroscopy that can be used for the infrared single cell characterization—in our case—*P. falciparum*-infected human RBCs. Potential advantages and limitations of the mentioned methods prompted us to carry out these experiments and to compare between them. Furthermore, if all three discussed techniques can deliver similar molecular information, our aim is to prioritize the most economical method in terms of cost of consumables, manpower and ease of operations and data processing.

All techniques used in this work can provide quantitative distribution of bio-molecules such as DNA, carbohydrates, proteins, lipids and in the case of *P. falciparum*, hemozoin. However,

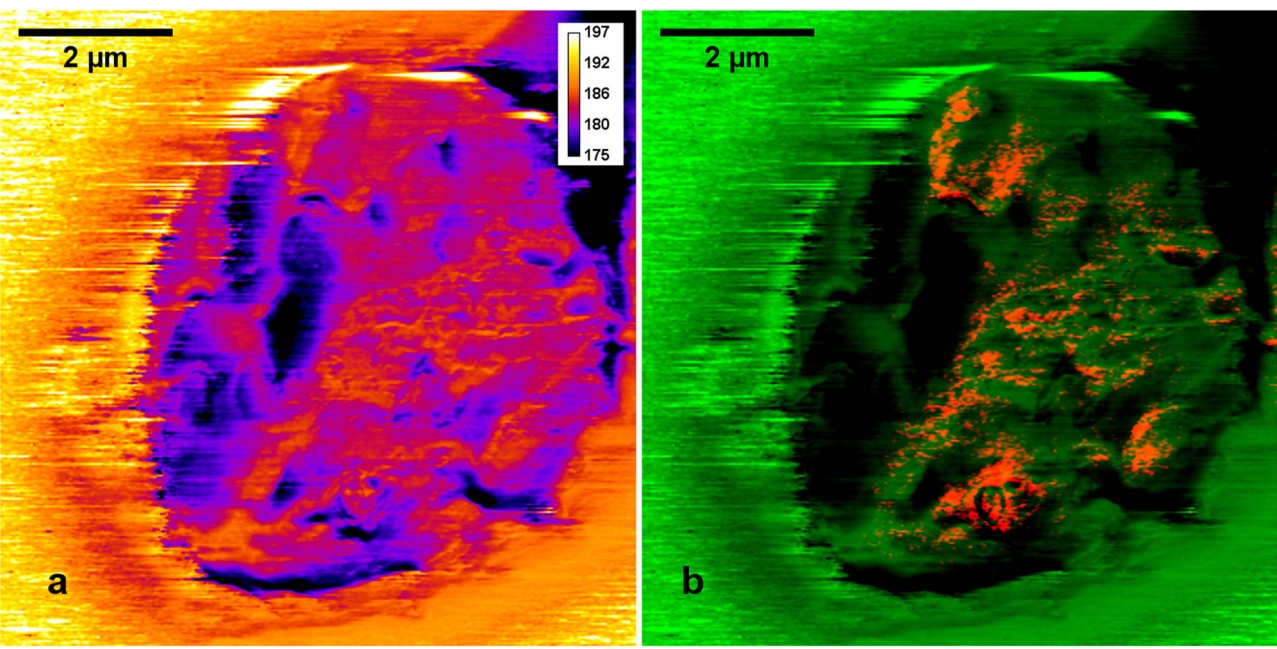

**Fig. 6 Co-mapping of local elastic properties with distribution of band characteristic for hematozoin presence in infected RBC.** PLL frequency channel recorded simultaneously during AFM-IR experiments (**a**), RGB composite image of PLL frequency (green channel) and 1740 $cm^{-1}$ (red channel) distributions (**b**).

**Table 1 Practical advantages and disadvantages of the spectroscopic methods observed from our analysis of individual *Plasmodium*-infected RBCs.**

| | FTIR microspectroscopy | O-PTIR spectroscopy | AFM-IR |
|---|---|---|---|
| Advantages | - Thousands of FTIR spectra are collected in the timescale of min (16 384 spectra; 8 $cm^{-1}$ resolution, 1024 scans were collected in 32 min)<br>- Source of IR (Globar or synchrotron radiation SR) emits with a power of ~80 µW over the whole spectral range | - Sub-micron resolution is suitable for analysis different area within single cell<br>- 1 spectrum (20 scans) with effective spectral resolution of 6 $cm^{-1}$ was collected within 3 min<br>- Chemical visualization can be performed with sub-micron resolution; tiny structures within the sample can be easily presented<br>- Collected spectra are free from spectral artifacts (e.g., Mie Scattering)<br>- Depending on the size and resolution of collected images; usually few minutes are needed | - Nanometer resolution enables for very precise analysis of heterogenous small object such as infected cell<br>- 1 spectrum (16 scans) with effective spectral resolution of 6 $cm^{-1}$ is collected within 3 min<br>- Chemical visualization can be performed with nanometer resolution; miniscule structures within the sample can be easily presented |
| Disadvantages | - Presence of artifacts like Mie scattering in collected spectra<br>- Diffraction-limited spatial resolution not suitable to analyse objects smaller than 5–15 µm; especially for single cell analysis<br>- Extremely large volumes of data generated and associated with chemical images (in the order of gigabytes of data per hour)<br>- Cryogenic cooling (liquid nitrogen) of the detectors; must be done by trained personnel | - Setting the proper power levels of IR and green (probe) light is needed prior to experiments (in our experiments, power of approximately 1–5 mW per spectrum was needed).<br>- Chemical visualization is done for one wavenumber at a time during one experiment; the process could be automated | - Time-consuming adjustments and optimization of the system<br>- Setting the proper power levels of IR is needed prior to experiments (in our experiments, 0.2–1 mW per spectrum was needed).<br>- Chemical visualization is done for one wavenumber at a time; the process could be automated<br>- Depending on the size—only one cell can be analyzed in reasonable time; usually tens of minutes are needed<br>- In contact mode operation, special attention must be paid to avoid sample damage<br>- Consumables: relatively expensive and exclusive AFM-IR designed tips are required for the experiments |

there are essential factors (summarized in Table 1) which determine the method's suitability for the analysis of the biochemical landscape of single cells.

Our investigation started with the analysis of single, healthy RBCs (control) as they are well characterized in terms of morphology and composition. Measurements done by FTIR microspectroscopy and O-PTIR spectroscopy show that only O-PTIR is able to reveal spectroscopic differences between the center vs. the periphery part of the single, healthy cell (Fig. 2). PCA performed on spectra collected for the control and infected RBCs presented in this work, demonstrates that FTIR microspectroscopy cannot separate these two groups. Clear differentiation is perceived for the spectra obtained by O-PTIR. This suggests that submicron resolution is indeed required if local biochemical variability within single cell must be checked.

In order to explore sub-cellular structures (especially those in nanometer scale range), on the other hand, AFM-IR seems to be valuable and irreplaceable tool providing deeper insight into chemical and viscoelastic properties with spatial resolution superior to any other techniques.

Based on the presented results, we conclude that O-PTIR and AFM-IR spectroscopies are preferred choices for analyzing complex samples, such as *Plasmodium*-infected blood cells, offering chemical information pertinent to a heterogeneous environment within a single cell.

The sub-micron and nanometer-resolution offered by these techniques make them ideal modalities for better understanding of malaria pathophysiology, where sample availability is limited as it is in the case of *P. vivax* infections. These techniques thus

open new avenues for non-invasive monitoring of biochemical processes occurring within single cells.

## Methods
All spectroscopic/imaging experiments were performed at the branch of the upgraded ISMI beamline (at the Singapore Synchrotron Light Source, National University of Singapore) by means of:

1. Hyperion 3000 (IR microscope) equipped with FPA detector (128 pixels by 128 pixels) attached to Vertex 80 v (both Bruker) spectrometer – for FTIR microspectroscopy,
2. mIRage microscope produced by the Photothermal Spectroscopy Corp (PSC) – for the O-PTIR sub-micron IR characterization,
3. nanoIR3 system produced by Bruker Nano – for nanoscale AFM-IR absorption spectroscopy.

**Preparation of *Plasmodium*-infected blood samples**. 3D7 strain of *P. falciparum* parasites grown in human blood (purchased from Interstate Blood Bank, USA) were used in all experiments, in agreement with approved protocols of Singapore University of Technology and Design. Synchrony of the cultures was achieved by frequent selection of ring-stage infections using sorbitol treatment. Trophozoite stage parasites (~30 hours post-invasion) were used in all experiments reported in this work.

Infected RBCs (around 5% parasitemia) were washed once in 1X PBS followed by fixation in 2% paraformaldehyde PFA for 15 min. Fixed cells were smeared on $CaF_2$ windows (Crystran, UK) and air-dried for further use. Samples were first examined under a 4x (NA 0.1), 10x (NA 0.25) and 40x (NA 0.65) objective lens (Leica DM750) and microscopic images were captured with a Leica ICC50W digital camera. Since the smears were not stained, the infected cells were recognized based on the typical black hemozoin dots of trophozoite stage parasites. Infected cells were selected and labeled from high contrast 40x microscopic images and the corresponding 10x and 4x images. Further scans by FTIR, O-PTIR or AFM-IR were focused on selected cells.

**Details of the experiments FTIR microspectroscopy.** Analysis of RBCs was performed using FTIR microspectroscopy in transmission mode with a Bruker Hyperion 3000 IR microscope with a FPA detector attached to the Vertex 80 v spectrometer. Since cells were placed on a $CaF_2$ window, a location adjacent to the sample at a clear, clean spot on that window was used for background signal collection.

An IR objective lens with 15× magnification was used that provides a pixel size equal to 2.7 $\mu m^2$ (with 128 by 128 pixels in each direction of the FPA detector). The number of co-added scans were tested in order to ensure sufficient signal-to-noise ratio. 1024 scans provided a balance between good quality spectra within a reasonable amount of time (32 mins).

A spectral range of 3845 to 900 $cm^{-1}$ was set with a spectral resolution of 8 $cm^{-1}$. The zero filling factor was set to 2 and a Blackman-Harris 3-Term apodization function with phase resolution of 32 and power phase correction mode was selected for converting measured interferograms to final spectra.

From a single experiment, 16,384 spectra were collected from an area of approximately $345 \times 345$ $\mu m^2$.

**Sub-micron IR characterization (O-PTIR).** O-PTIR sub-micron IR character-ization was carried out using a mIRage microscope (PSC, USA). As a source of IR radiation, a tuneable, pulsed mid–IR Quantum Cascade Laser (QCL) was used, operating with 300 ns pulses at a repetition rate of up to 100 kHz. The selected area within the sample was illuminated, which caused small photothermal perturbations in the form of rapid conversion of photon energy into heat, lasting for a few microseconds[31]. These perturbations can be probed by a green (532 nm) laser, the position of which was co-located with the IR light. Here, the visible probe beam detects localized reflectivity changes on the sample surface, hence O-PTIR's sensitivity was applicable only to the materials' IR absorptivity at a given wavenumber. The laser power levels of the IR and probe were each set to approximately 1–5 mW per spectrum (wavelength dependent). O-PTIR measurements were done in a non-contact reflection mode while providing FTIR transmission-like infrared spectra that were artifact-free (without Mie scattering and other effects). Available at the ISMI beamline, QCL comprises 4 chips and enables spectra collection within the mid-IR fingerprint region of 1800–900 $cm^{-1}$. The number of co-added scans per spectrum was 20 to maintain a good signal to noise ratio. The effective spectral resolution was approximately 6 $cm^{-1}$ (with 2 $cm^{-1}$ data point spacing).

The spatial resolution in the O-PTIR method is dependent on the spot size of the short wavelength green laser. During experiments, its size was maintained below 1 $\mu m$ to achieve submicron spatial resolution.

The background signal, by means of measuring photothermal perturbations returned from a reflective surface, was recorded to take into account the influence of the IR laser power distribution on the final spectrum. Before collecting O-PTIR spectra from all samples, the system was wavenumber-calibrated using a manufacturer-supplied test sample composed of a cross-section of PMMA and PS beads for the 1732 $cm^{-1}$ and 1601 $cm^{-1}$ bands.

Prior to sample analysis, microscopic images of cells were collected with different objectives: low magnification visible (refractive objective lens) 10x, 0.3NA (working distance of about 15 mm) and high magnification (reflective Cassegrain objective lens) 40x, 0.78NA (working distance of about 8 mm). The latter was also

used for data (spectra and single wavenumber images) collection, which were saved and processed in PTIR Studio 4.0 software supplied with the instrument.

Samples (RBCs on $CaF_2$ disks) to be analyzed were placed on the automated scanning stage in the sample chamber; the entire instrument was purged with dry nitrogen during all experiments in order to eliminate water vapor absorption.

Apart from single spectra taken from chosen areas within the selected cells, images for particular wavenumbers and hyperspectral data were also collected.

The hyperspectral approach is very similar to chemical mapping experiments performed by a conventional IR microscope with a single-element detector, where the aperture size and the region of interest are selected by horizontal and vertical slits. Steps in the $x$ and $y$ axis are set and raster scanning is carried out (a single spectrum is collected from one point, then the sample is moved to the next point where next spectrum is taken).

In O-PTIR spectroscopy, there is no aperture to be set, the size of the hyperspectral object, its spacing in the $x$ and $y$ direction is selected by the analyst. A hyperspectral experiment is performed automatically, prior to every single spectrum collection the autofocus function selects the best position in $z$ direction for an analyzed spot for the highest amplitude of reflection signal at a given wavenumber, e.g., 1452 $cm^{-1}$.

**Nanoscale IR Characterization (AFM-IR experiments).** A $CaF_2$ window with RBCs fixed on it, was mounted on a steel holder and analyzed within the 1800 – 900 $cm^{-1}$ with a resolution of about 6 $cm^{-1}$ (spectral point density equal to 2 $cm^{-1}$) by means of the nanoIR3 instrument (Bruker Nano, USA); Analysis Studio 3.16 provided with the instrument was used to collect all experimental data. The same QCL laser (described earlier) was used as the source of IR. All the measurements were performed using a gold-coated AFM tip with an apex of nominal curvature radius of sub-30 nm (Anasys Instruments, USA). The cantilever had a nominal spring constant of 0.07−0.4 N/m and a resonance frequency of 13 ± 4 kHz, while in a "free-state." Rapid laser pulses (QCL) tuned to a corresponding absorption bands occurring in the samples, caused rapid thermal expansions thereby resulting in an oscillation of the AFM tip placed in contact with the sample. The obtained spectrum was the amplitude of the cantilever oscillation, which is proportional to the absorption coefficient of the sample at different wavenumber values after normalizing against the background[24].

Prior to experiments, the nanoIR3 system was calibrated and optimized using a test sample (the same as for O-PTIR technique) provided by the manufacturer. The background was collected to compensate the input from the atmospheric gases and to take into account the power distribution of the IR laser delivered on the sample.

For nanoscale IR data collection, the instrument is operated in resonance enhanced AFM-IR mode where the laser pulse repetition rate is synced to the 2nd eigenmode of the cantilever contact resonance. This cantilever oscillation signal is detected in the position sensitive photodetector (PSPD) of the AFM and fed to an internal lock-in amplifier to extract the demodulated amplitude signal proportional to IR absorption. For each spectrum, 16 scans were co-averaged. For chemical imaging, the scanning was performed with 400 points in the $x$ and 400 points in the $y$ direction at 0.5 Hz scan rate.

---

**Table 2 Summary of all spectroscopic experiments performed on *Plasmodium*-infected RBC samples.**

| Method | FTIR microspectroscopy (Hyperion 3000) | O-PTIR (mIRage) | AFM – IR (nanoIR3) |
|---|---|---|---|
| Spectral range | 3845 to 900 $cm^{-1}$ | 1800−900 $cm^{-1}$ | 1800−900 $cm^{-1}$ |
| Spatial resolution | IR wavelength dependent; theoretical limit can be calculated using Rayleigh criterion: 15 $\mu m$ (at 1000 $cm^{-1}$), 9.2 $\mu m$ (at 1650 $cm^{-1}$), and 5.4 $\mu m$ (at 2800 $cm^{-1}$) | <1 $\mu m$ resolution; theoretical limit can be calculated using Rayleigh criterion: 416 nm with 532 nm probe laser | ~40 nm resolution; theoretical limit depends on the AFM tip radius, which is ~40 nm |
| Single spectrum | 128 ×128 = 16384 spectra collected in each experiment (FPA detector) | Collected from selected spots | Collected from selected spots |
| IR absorption images = Spatial distributions of absorption for selected wavenumbers | Done as a post-processing (integrals calculation) of collected array of the spectra | Done as a post-processing (integrals calculation) of collected hyperspectral data or Collected for selected wavenumbers; one image = chemical distribution for one wavenumber | Collected for selected wavenumbers; one image = chemical distribution for one wavenumber |
| Topography information | Not available | Collected in the form of mIRage laser reflectivity image (the relative reflectivity of the green laser) | Collected in form of AFM images (height channel) |
| Additional information | Not available | Not available | Viscoelastic properties of the sample by tracking PLL frequency changes |

With all IR absorption images (for selected wavenumbers), not only AFM images (height signal) were simultaneously recorded for the same regions of interest in order to gain information regarding the topography of the sample, but also the contact resonance frequency of the cantilever using the Phase Locked Loop (PLL) functionality to gain insight into the nanomechanical sample properties.

As described earlier, these methods provide the unique opportunity to analyse exactly the same sample as all these methods can be treated as non-destructive; after every experiment the specimen was left in pristine condition.

Table 2 presents the summary of experiments carried out on single RBC by means of three modalities discussed earlier.

**Reporting summary**. Further information on research design is available in the Nature Research Reporting Summary linked to this article.

## Data availability

The datasets generated during the current study are available from the corresponding author on reasonable request.

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

## Acknowledgements

The authors would like to acknowledge the Singapore Synchrotron Light Source (SSLS) for providing the facility necessary for conducting the research. The Laboratory is a National Research Infrastructure under the National Research Foundation Singapore. T.C., R.N. and R.C. acknowledges the infrastcructure and laboratory support from the Singapore University of Technology & Design (SUTD). R.C. was funded through the following grants: T1MOE1702 (MOE Tier 1 Grant through the Singapore University of Technology & Design) and RGUOO180301 (Marsden Grant Sub-award through the University of Otago).

## Author contributions

A.B.: contributed to study design, performed all experiments, analyzed data and wrote the manuscript; KB: Contributed to study design, performed all experiments, analyzed data and wrote the manuscript; T.T.T.C.: contributed to study design, performed Plasmodium experiments, contributed to data analysis and assisted with manuscript preparation; R.N.: performed Plasmodium experiments, contributed to data analysis and assisted with manuscript preparation; P.E.H.: contributed to study design and provided reagents and analysis tools; R.A.: contributed to study design and provided reagents and analysis tools; M.L.: assisted with data analysis and verification; reviewed and edited the manuscript; M.K.: assisted with data analysis and verification; reviewed and edited the manuscript; A.R.: assisted with data analysis and verification, reviewed and edited the manuscript; R.C.: designed the study, funding and resource acquisition, verified the results, coordinated project, reviewed and edited the manuscript; M.B.: funding and resource acquisition, verified the results, reviewed and edited the manuscript.

## Competing interests

The authors declare no competing interests.
