## [Peer Review File · Communications Chemistry]

Reviewers' comments:

Reviewer #1 (Remarks to the Author):

Banas et al. present a study in which optical photothermal infrared (O-PTIR) spectroscopy and atomic force microscopy combined with infrared spectroscopy (AFM-IR) were systematically compared for the study of malaria. The paper is well motivated and clearly written. The results are free from obvious mistakes, and will have impacts for various research applications related to the infectious diseases. In principle, I strongly support the publication of the manuscript. However, the following minor comments and suggestions should be addressed prior to publication.

- In the introduction, Raman microscopy was mentioned as label-free techniques. I suggest including quantitative phase imaging techniques and comparing them with other label-free imaging techniques. Here are a few references:

-- Chandramohanadas, Rajesh, et al. "Biophysics of malarial parasite exit from infected erythrocytes." *PloS one* 6.6 (2011): e20869.

-- Park, YongKeun, Christian Depeursinge, and Gabriel Popescu. "Quantitative phase imaging in biomedicine." *Nature photonics* 12.10 (2018): 578-589.

-- Hayakawa, Eri H., et al. "Real-time cholesterol sorting in Plasmodium falciparum-erythrocytes as revealed by 3D label-free imaging." *Scientific Reports* 10.1 (2020): 1-12.

- In pages 4-5, the term "15x objective" should be corrected as "an objective lens." Also, throughout the manuscript, "an objective lens" was expressed as "objective," which should be corrected.

- In page 11 line 311, please provide the model number and the value for the numerical aperture.

- It was difficult to understand the results in Fig. 4. Can the authors elaborate the explanations in both the main text and the figure?

- In page 17, lines 423-428 seem a mistake, which should be removed.

- There are many minor English usage/syntax problems -- too many to list here. It's generally clear what the authors are trying to get across, but the errors are distracting. To make sure that the paper and its results are accessible to the majority of readers, the paper should be revised to conform to standard English grammar.

Reviewer #2 (Remarks to the Author):

The author of the paper "Do we really need sub-micron resolution to analyze single cell molecular features through vibrational spectroscopy? A pilot study using Plasmodium falciparum-infected Human Erythrocytes" compare three commercial infrared imaging systems (FTIR-FPA, AFM-IR and O-PTIR) by applying them to the same sample. In general, the authors demonstrate competence in using these techniques and the comparison itself is likely interesting to other researchers in the field. The basic premise of the work is certainly worthwhile and should be of interest to Comms Chem readers. The authors point of reference, the FPA-FTIR imaging system has been the gold standard in mid-IR imaging for a long time, while both O-PTIR and AFM-IR are more recent developments that promise to outperform the legacy technique when it comes to spatial resolution. I was thus quite interested in reading and reviewing this work. However, unfortunately, I have to say that the authors fall short of really providing a meaningful comparison. The manuscript should not be published in its current form.

While I point out specific issues with the paper in the following paragraphs, these only constitute the most egregious problems. Only fixing these issues therefore does not mean that I would consider the manuscript to be adept for publication. To get to that point, the authors would need to restate their research objective in a quantitative way such that the comparison between methods can be carried out in a meaningful way.

Tellingly, the issues with this paper start at the title. The authors seem to have chosen the title of their manuscript mainly because it sounds catchy. They ask the question "Do we really need sub-micron resolution to analyze single cell molecular features through vibrational spectroscopy?" After reading the manuscript several times I not only have no clue about their answer but also have no idea how they propose to go about finding said answer in the first place. In fact, they specifically state that they won't even try to understand their data with regard to this - in my opinion- central aspect of the investigation (line 152): "Herein, we deliberately refrain from any biochemical interpretation of the obtained results." That begs the question, why a sample that was likely hard to obtain and is relatively complex was chosen for the comparison, rather than a much simpler one such as a polymer blend or size reference beads. Which not only would have been easier to obtain, but also would have allowed to gain quantitative insights in the strengths and weaknesses of all three techniques.

The points of comparison between methods are all over the place. Specifically, the ability for "sparseness" in the wavelength domain is not necessarily an advantage for the complex type of sample the authors are measuring here. Especially for spectroscopy of the amid I band, single wavelength information is typically insufficient; shape, positions and relative intensities also matter. Hence, conventionally researchers that tackle complex bio-samples with EC-QCL spectroscopy will either collect broad band spectra or determine the most important wavelengths in the first step of their experiment (see works by the Baker and Bhargava groups for examples). Similarly, sparseness in sampling the spatial domain can be an advantage but it typically comes at the trade-off of positioning errors and challenges in aligning successive images (in AFM-IR that is certainly the case, in O-PTIR likely to a lesser degree, but still). Discussing this aspect could be a valuable part of the manuscript.

Furthermore, one aspect that is completely left out of the manuscript is that the three methods have vastly different methods of signal generation. Only FPA-FTIR here actually provides "true" absorption spectra that follow Beer's law (given the absence of other optical effects, such as Mie scattering artefacts), while O-PTIR and AFM-IR both have complex signal transduction chains that include optical parameters, but also thermal and mechanical sample properties. This aspect is briefly hinted at in the manuscript but then neglected. In my opinion, addressing this aspect is a crucial point for a direct comparison of methods. O-PTIR and AFM-IR might see more lateral variations in their signal, but those might either stem from variations in absorption or in one of the other parameters that go into the signal transduction chain.

The claim that AFM-IR is more "surface specific" (line 442) is incorrect. According to state of the art understanding of AFM-IR signal generation, the signal for the cells measured here should show contributions from the whole vertical pillar below the AFM tip, averaged in z. Recently, some works (e.g. by Quaroni) have demonstrated that higher laser repetition rates can reduce the information depth somewhat, but not to the point where it is significantly different in height from the volume sampled by O-PTIR and in any case such effects would not occur at the frequencies used in this work. What the researchers might be seeing instead in their AFM-IR images are variations of tip sample

contact stiffness across the surface (or topography cross talk). Here, too, a work from the Bhargava group (Kenkel et al) recently has demonstrated that better control of the AFM-IR instrument can remove these effects.

In addition to resorting to qualitative descriptions (e.g. "good signal to noise") rather than quantitative ones, and in parts not performing a comparison of techniques but rather of instruments and user interfaces (which is, in my opinion, inappropriate. For example, a vendor's decision not to include a specific visualization in their software package, doesn't constitute draw back of a technique itself) there are quite a few typos in the text. For example, the phrase "colour-coded" in fig 3 seems fishy.

Reviewer #3 (Remarks to the Author):

In the manuscript, the authors mainly focused on comparing three IR techniques (FTIR, O-PTIR, and AFM-IR) on the microscopy/spectroscopy measurements of red blood cells. Overall, the content is organized and straightforward. I would recommend a major revision with the following questions/concerns to be addressed.

- 1) At the beginning of the manuscript, the authors gave a detailed introduction of malaria parasites and elucidated the necessity of identifying their unique biochemical signatures. However, in the Results section, only Amide I mapping is demonstrated, which is a universal component of cells and does not contain much useful information. Although the authors said further data analysis is beyond the scope of this manuscript, the background introduction still seems an over-claim.
- 2) As this manuscript's primary goal is to compare three IR methods, more technique details should be included. For example, it would be beneficial to introduce the basic working principles of these methods and show a schematic illustration of setups.
- 3) More information can be included in Table 1, such as image acquisition speed and laser power required. AFM-IR is supposed to benefit from the field enhancement by the metallic tip. So, is the IR power much smaller than O-PTIR or FTIR?
- 4) In Figure 2f, the height color bar is missing. One major advantage of AFM is the capability of extracting height information of samples. From this image, one can see the surface of the cell is relatively rough. Could this be part of the reason for the observed different protein distribution in Figures 5c and 5d? Considering a soft tip was used whose oscillations might be sensitive to the sample's sharp features, it is likely the different distributions come from artifacts. Also, the cell on the right side in Figure 5d shows strong responses on its edge. What could be the reason for this?
- 5) In lines 366 and 367, the authors estimated the area where O-PTIR and AFM-IR spectra were taken. 500 nm and 40 nm are the length or diameter of the area, and they should not be directly used as 500 nm² or 40 nm².
- 6) This is trivial, but the authors might want to improve their formatting and figure qualities. Some figures and labels are not aligned.

Reviewer # 1

Banas et al. present a study in which optical photothermal infrared (O-PTIR) spectroscopy and atomic force microscopy combined with infrared spectroscopy (AFM-IR) were systematically compared for the study of malaria. The paper is well motivated and clearly written. The results are free from obvious mistakes, and will have impacts for various research applications related to the infectious diseases. In principle, I strongly support the publication of the manuscript. However, the following minor comments and suggestions should be addressed prior to publication.

We thank the Reviewer for the valuable comments and appreciation of the work.

Comment 1: In the introduction, Raman microscopy was mentioned as label-free techniques. I suggest including quantitative phase imaging techniques and comparing them with other label-free imaging techniques. Here are a few references:

- Chandramohanadas, Rajesh, et al. "Biophysics of malarial parasite exit from infected erythrocytes." *PloS one* 6.6 (2011): e20869.
- Park, YongKeun, Christian Depeursinge, and Gabriel Popescu. "Quantitative phase imaging in biomedicine." *Nature photonics* 12.10 (2018): 578-589.
- Hayakawa, Eri H., et al. "Real-time cholesterol sorting in Plasmodium falciparum-erythrocytes as revealed by 3D label-free imaging." *Scientific Reports* 10.1 (2020): 1-12.

Response: We thank the Reviewer for the suggestion. These references are now included in the revised manuscript and comparisons are made between label-free techniques described in these references, as described below.

Label-free techniques such as Raman imaging microscopy has been tested as a potential method to diagnose plasmodium infection of human RBCs on the basis of strong scattering from the hemozoin pigment [13]. However, a rather low signal to noise ratio, high autofluorescence and potential photodamage limits its effectiveness in revealing molecular structures of infected single RBCs.

Other label-free methods such as tomographic phase microscopy (TPM) were also previously explored to study plasmodium infected RBCs as it allows for visualization of intracellular organelles, membranous structures and life-stage transitions from 3D refractive index tomograms [14-16].

14. Chandramohanadas, R. *et al.* Biophysics of malarial parasite exit from infected erythrocytes. *PLoS One* **6**, (2011).
15. Park, Y. K., Depeursinge, C. & Popescu, G. Quantitative phase imaging in biomedicine. *Nat. Photonics* **12**, 578–589 (2018).
16. Hayakawa, E. H., Yamaguchi, K., Mori, M. & Nardone, G. Real-time cholesterol sorting in Plasmodium falciparum-erythrocytes as revealed by 3D label-free imaging. *Sci. Rep.* **10**, 1–13 (2020).

Comment 2: In pages 4-5, the term “15x objective” should be corrected as “an objective

lens.” Also, throughout the manuscript, “an objective lens” was expressed as “objective,” which should be corrected.

Response: Thanks for pointing this out, we have re-phrased it throughout the manuscript.

Comment 3: In page 11 line 311, please provide the model number and the value for the numerical aperture.

Response: We have included the necessary details as suggested to the section:

Preparation of plasmodium-infected blood samples:

Samples were first examined under 4x (NA 0.1), 10x (NA 0.25) and 40x (NA 0.65) objective lens (Leica DM750) and microscopic images were captured by Leica ICC50W digital camera.

Comment 4: It was difficult to understand the results in Fig. 4. Can the authors elaborate the explanations in both the main text and the figure?

Response: We thank the Reviewer for pointing this out, we have made significant changes to the figures to improve clarity in this revised submission.

Fig. 4 (in first version of our paper), depicts the dendrogram - the result of hierarchical cluster analysis (HCA). HCA was used to adequately illustrate the differences, if any, among the collected spectra. Prior to HCA, all spectra were normalized (min-max normalization), otherwise, it would be difficult or even impossible to calculate the differences between each pair of the spectra.

As can be seen in the previously submitted version, the dendrogram in Fig. 4 clearly shows a hierarchical relationship between the spectra that were chosen for comparison. Spectra were allocated by the algorithm into three groups according to the type of experiments they were part of. FTIR spectra are linked together at very low height that implicates high correlation among them (or similarity). For O-PTIR and AFM-IR spectra, prominent differences are noticed since representative spectra are joined together further apart in comparison to FTIR spectra.

However, as was suggested by the Reviewers and Editor, the Results section has been modified and in the current version results of HCA are not presented.

Comment 5: In page 17, lines 423-428 seem a mistake, which should be removed.

Response: We have corrected this mistake.

Comment 6: There are many minor English usage/syntax problems -- too many to list here. It's generally clear what the authors are trying to get across, but the errors are distracting. To make sure that the paper and its results are accessible to the majority of readers, the paper should be revised to conform to standard English grammar.

Response: We have spent considerable time in improving the manuscript and eliminate errors, in view of the Reviewer's comment. Changes made are indicated in track changes mode for easy reference.

Reviewer #2

Comment 1: The author of the paper "Do we really need sub-micron resolution to analyze single cell molecular features through vibrational spectroscopy? A pilot study using *Plasmodium falciparum*-infected Human Erythrocytes" compare three commercial infrared imaging systems (FTIR-FPA, AFM-IR and O-PTIR) by applying them to the same sample. In general, the authors demonstrate competence in using these techniques and the comparison itself is likely interesting to other researchers in the field. The basic premise of the work is certainly worthwhile and should be of interest to Comms Chem readers. The authors point of reference, the FPA-FTIR imaging system has been the gold standard in mid-IR imaging for a long time, while both O-PTIR and AFM-IR are more recent developments that promise to outperform the legacy technique when it comes to spatial resolution. I was thus quite interested in reading and reviewing this work. However, unfortunately, I have to say that the authors fall short of really providing a meaningful comparison. The manuscript should not be published in its current form.

While I point out specific issues with the paper in the following paragraphs, these only constitute the most egregious problems. Only fixing these issues therefore does not mean that I would consider the manuscript to be adept for publication. To get to that point, the authors would need to restate their research objective in a quantitative way such that the comparison between methods can be carried out in a meaningful way.

Response: We thank the Reviewer for her/his comment and have re-phrased our objectives, and more quantitative comparisons are made as suggested.

We have now made extensive changes to the Results section as listed below:

(i) Inclusion of data on comparable spectral data collection using a reference material

To confirm the equivalence for IR spectra collection between FTIR microspectroscopy, O-PTIR and AFM-IR, similarity of resultant spectra using a reference sample was first checked (using a microtomed section of a cured epoxy adhesive (~800 nm) placed CaF₂). This data is now included as Fig. 1. Only after this validation, we proceeded with comparative analysis of spectra arising from RBCs or *Plasmodium*-infected RBCs reported in this paper.

(ii) Detailed analysis of individual RBC (control) by means of FTIR microscopy and O-PTIR

The spatial distribution of cellular constituents: amide I (C=O and C-N) at ~1650 cm⁻¹ and proteins/lipids (νCOO₂ of fatty acids and amino acid side chains) at ~1391 cm⁻¹ have been presented. To better compare, FTIR (experiments with FPA detector) and O-PTIR results, O-PTIR hyperspectral data is being discussed. Hyperspectral approach is very similar to chemical mapping experiments performed by means of conventional IR microscope with a

single-element detector where aperture size and the region of interest are selected by the set of horizontal and vertical slits, steps in x and y axis are set and raster scan is carried out (single spectrum is collected from one point, then the sample is moved to the next point, where next spectrum is taken),

Comparison of single spectra:

- FTIR microspectroscopy- one experiment with the usage of FPA (128x128) produces a total of 16,384 spectra, the cell of interest is covered by approximately 9 elements of the FPA (theoretical size of the single pixel is 2.7 by 2.7 μm^2), some of them extending beyond the cell area into the substrate, for the comparison with O-PTIR results, three spectra were chosen from the middle part of the selected cell;
- for the same cell, 169 spectra (hyperspectral experiment) were collected with a spacing of 0.5 μm in x and y directions in case of O-PTIR spectroscopy; five of the six presented spectra reveal the biochemistry of the area smaller than the size of single detector of FPA, but some obvious variations in the positions of amide I and the intensities of other bands related to other cell constituents are observed. Spectrum 6 was recorded from the empty region 0.5 μm away without cell presence; which is a remarkable photothermal infrared response providing for direct evidence of the submicron spatial resolution capability with O-PTIR spectroscopy.

(iii) Comparison of results collected for 20 infected and 20 control RBCs (FTIR microspectroscopy vs O-PTIR)

It is difficult to conclude whether lack/or presence of visible differences among spectra collected solely for the same single cell (control RBC) can be treated as a general trend for all analyzed cells. Therefore, broader perspectives are needed to draw reliable conclusions. Hence, analysis performed on: 124 spectra collected from uninfected (control) and 97 spectra from infected RBCs (iRBC - the trophozoite phase) in case of FTIR microspectroscopy, for O-PTIR spectroscopy 64 spectra for control and 164 spectra for iRBC were taken into account:

- PCA was done to evaluate any clustering patterns among studied spectra;
- comparison of O-PTIR mean spectra collected for control and infected RBCs with the PC1 loading was done in order to find ROI in wavenumbers which could play the role in observed clustering for O-PTIR measurements;
- comparison of area calculated under the spectrum line within the region 1725-1700 cm^{-1} for each spectrum collected for control and infected RBCs by means of FTIR microspectroscopy and O-PTIR was done in order to see if the presence of Hz can be found in spectra collected for infected RBCs;
- comparison of the relative positions of the amide I band in single spectra collected for control and infected RBCs by means of FTIR microspectroscopy and O-PTIR was done to check if there is any shift which could point for modification in the secondary structure of proteins composition.

(iv) AFM experiments (example of results for single infected RBC is presented)

For AFM-IR experiments results - RGB composite image (red channel - topography, green channel -1660 cm^{-1} , blue channel - 1740 cm^{-1}) are presented to show an advantage of spatial resolution over FTIR and O-PTIR:

- single spectra were highlighted to show the dynamic biochemistry not only within amide I and II bands but also within the regions assigned to nucleic acids (including the symmetric ($\sim 1080\text{ cm}^{-1}$) and asymmetric ($\sim 1230\text{ cm}^{-1}$) phosphodiester vibration) and amide III band ($\sim 1286\text{ cm}^{-1}$)

-additional PLL (Phase Locked Loop) frequency spatial distribution was presented to show that stiffness within infected RBC may be connected to the area with traces of hemozoin.

Tellingly, the issues with this paper start at the title. The authors seem to have chosen the title of their manuscript mainly because it sounds catchy. They ask the question "Do we really need sub-micron resolution to analyze single cell molecular features through vibrational spectroscopy?" After reading the manuscript several times I not only have no clue about their answer but also have no idea how they propose to go about finding said answer in the first place. In fact, they specifically state that they won't even try to understand their data with regard to this - in my opinion- central aspect of the investigation (line 152): "Herein, we deliberately refrain from any biochemical interpretation of the obtained results." That begs the question, why a sample that was likely hard to obtain and is relatively complex was chosen for the comparison, rather than a much simpler one such as a polymer blend or size reference beads. Which not only would have been easier to obtain, but also would have allowed to gain quantitative insights in the strengths and weaknesses of all three techniques.

Response: Selection of human RBCs and plasmodium infected RBCs for this comparative analysis was intentional, motivated by sample's range of complexities (healthy vs infectious), relevance to medicine and availability of samples through existing network of collaborations.

By choosing human blood cells and malaria infected cells for this work, we were able to investigate how these three techniques can derive important information on a relatively simple living cell (RBC with no nucleus and internal organelles provides an easy, accessible cell type) and a more complex cell (iRBCs harbouring malaria parasites- changes arising from infection and parasite development within adding in extra features of a typical eukaryotic cell).

The points of comparison between methods are all over the place. Specifically, the ability for "sparseness" in the wavelength domain is not necessarily an advantage for the complex type of sample the authors are measuring here. Especially for spectroscopy of the amide I band, single wavelength information is typically insufficient; shape, positions and relative intensities also matter. Hence, conventionally researchers that tackle complex bio-samples with EC-QCL spectroscopy will either collect broad band spectra or determine the most important wavelengths in the first step of their experiment (see works by the Baker and Bhargava groups for examples). Similarly, sparseness in sampling the spatial domain can be an advantage but it typically comes at the trade-off of positioning errors and challenges in aligning successive images (in AFM-IR that is certainly the case, in O-PTIR likely to a lesser degree, but still). Discussing this aspect could be a valuable part of the manuscript.

Response: We thank the Reviewer for this comment. The discussion section have been extensively revised to improve its clarity.

To avoid comparison of IR absorption images collected only for one selected wavenumber, hyperspectral data sets were taken into account for O-PTIR spectroscopy.

Spatial distributions of absorption for selected wavenumbers have been calculated as an integral of characteristic bands characteristic, for both FTIR microspectroscopy and O-PTIR spectroscopy.

We have not experienced frequent issues with positioning errors and aligning successive images while using our nanoIR3 system, hence, it was not discussed in the paper. Based on our experience with work on nanoIR2 and nanoIR3, we have to stress that the latter system have been successfully improved in terms of stability, repeatability and noise-resistance.

Furthermore, one aspect that is completely left out of the manuscript is that the three methods have vastly different methods of signal generation. Only FPA-FTIR here actually provides "true" absorption spectra that follow Beer's law (given the absence of other optical effects, such as Mie scattering artefacts), while O-PTIR and AFM-IR both have complex signal transduction chains that include optical parameters, but also thermal and mechanical sample properties. This aspect is briefly hinted at in the manuscript but then neglected. In my opinion, addressing this aspect is a crucial point for a direct comparison of methods. O-PTIR and AFM-IR might see more lateral variations in their signal, but those might either stem from variations in absorption or in one of the other parameters that go into the signal transduction chain.

Response: We thank the Reviewer for this comment.

The suggested details have been included at the beginning of Results section in this revised version. To prove the equivalence of three methods for IR spectra collection, prior to comparison of the results obtained for RBCs, the similarity of resultant spectra collected during experiments by means of FTIR microspectroscopy, O-PTIR and AFM-IR was checked. As a reference sample, a microtomed section of a cured epoxy adhesive (thickness of ~800 nm) placed on CaF₂ window was used

The claim that AFM-IR is more "surface specific" (line 442) is incorrect. According to state of the art understanding of AFM-IR signal generation, the signal for the cells measured here should show contributions from the whole vertical pillar below the AFM tip, averaged in z. Recently, some works (e.g. by Quaroni) have demonstrated that higher laser repetition rates can reduce the information depth somewhat, but not to the point where it is significantly different in height from the volume sampled by O-PTIR and in any case such effects would not occur at the frequencies used in this work. What the researchers might be seeing instead in their AFM-IR images are variations of tip sample contact stiffness across the surface (or topography cross talk). Here, too, a work from the Bhargava group (Kenkel et al) recently has demonstrated that better control of the AFM-IR instrument can remove these effects.

Response: We thank the Reviewer for this comment. Based on Reviewers' comments, the Results section have been completely modified. The phrase "surface specific" have been removed. In current version example of two selected for iRBC bands distributions: amide I (at ~1660 cm⁻¹) and stretching vibrations of C=O in lipids, which can contribute to Hz

formation (at 1740 cm^{-1}) are presented. 1740 cm^{-1} was selected as a way of finding a presence of hemozoin traces. We are aware that presenting only single distribution of selected bands are not entirely meaningful especially for the complex sample; to collect AFM-IR chemical image only one wavenumber is chosen, information about changes in its shape, shifts in position are neglected. However, these images can be very useful in finding potentially interesting area to collect spectra that can be used for further analysis. As spatial resolution for AFM-IR ($\sim 40\text{ nm}$) means that to cover the area $1\text{ by }1\text{ }\mu\text{m}^2$, 625 spectra should be collected, knowing the precise region of interest for spectra collection is undoubtedly an advantage.

Additionally, PLL frequency distribution was presented in the current version of our paper. As the probe scans across the sample surface, the contact resonance of the probe changes, due to the stiffness differences between various sample components, differences in the contact area and force interaction between the tip and the sample. To collect meaningful data, measurements were done in PLL mode (phase-locked loop). Phase of the signal was fed to the PLL to track contact resonance frequency, hence all the nanomechanical info is manifested in the PLL frequency channel.

In addition to resorting to qualitative descriptions (e.g. "good signal to noise") rather than quantitative ones, and in parts not performing a comparison of techniques but rather of instruments and user interfaces (which is, in my opinion, inappropriate. For example, a vendor's decision not to include a specific visualization in their software package, doesn't constitute draw back of a technique itself) there are quite a few typos in the text. For example, the phrase "colour-codded" in fig 3 seems fishy.

Response: We thank the Reviewer for this comment in view of which more quantitative comparisons are made in this revised manuscript. It was not our intention to compare instruments and user interfaces in our manuscript, it was not done on purpose, our aim was to compare all advantages and disadvantages of the techniques from the Users' point of view.. We hope that current revised version is free of such unintentional comparisons.

We have also spent considerable time to minimize typos etc in this revised submission.

Reviewer #3 (Remarks to the Author):

In the manuscript, the authors mainly focused on comparing three IR techniques (FTIR, O-PTIR, and AFM-IR) on the microscopy/spectroscopy measurements of red blood cells. Overall, the content is organized and straightforward. I would recommend a major revision with the following questions/concerns to be addressed.

1) At the beginning of the manuscript, the authors gave a detailed introduction of malaria parasites and elucidated the necessity of identifying their unique biochemical signatures. However, in the Results section, only Amide \square mapping is demonstrated, which is a universal component of cells and does not contain much useful information. Although the

authors said further data analysis is beyond the scope of this manuscript, the background introduction still seems an over-claim.

Response: We chose human blood cells and malaria infected cells for this work, as a way to compare how these three techniques can derive important information on a relatively simple living cell (RBC with no nucleus and internal organelles provides an easy, accessible cell type) and a more complex cell (RBCs harbouring malaria parasites- changes arising from infection adding in extra features of a typical eukaryotic cell).

Based on Reviewers' comments, the Results section have been completely modified.

Results section has been completely changed:

- to prove the equivalence of FTIR microspectroscopy, O-PTIR and AFM-IR for IR spectra collection, prior to comparison of the results obtained for RBCs, the similarity of resultant spectra was checked. As a reference sample, the cross section of epoxy slice (thickness of ~800 nm) placed on CaF₂ was used,

Analysis of selected single RBC (control) by means of FTIR microscopy and O-PTIR

- the spatial distribution of cellular constituents: amide I (C=O and C-N) at ~1650 cm⁻¹ and proteins/lipids (νCOO₂ of fatty acids and amino acid side chains) at ~1391 cm⁻¹ have been presented. To better compare, FTIR (experiments with FPA detector) and O-PTIR hyperspectral data is being discussed. Hyperspectral approach is very similar to chemical mapping experiments performed by means of conventional IR microscope with a single-element detector where aperture size and the region of interest are selected by the set of horizontal and vertical slits, steps in x and y axis are set and raster scan is carried out (single spectrum is collected from one point, then the sample is moved to the next point, where next spectrum is taken),

-comparison of single spectra:

- FTIR microspectroscopy- one experiment with the usage of FPA (128x128) produces 16384 spectra, the cell of interest is covered by approximately 9 elements of the FPA (theoretical size of the single pixel is 2.7 by 2.7 μm²), some of them extending beyond the cell area into the substrate, for comparison with o-PTIR results, three spectra were taken from the middle part of the selected cell;
- for the same cell, 169 spectra (hyperspectral experiment) were collected with a spacing of 0.5 μm in x and y directions in case of O-PTIR spectroscopy; five of the six presented spectra reveal the biochemistry of the area smaller than the size of single detector of FPA, but some obvious variations in the positions of amide I and the intensities of other bands related to other cell constituents are observed. Spectrum 6 was recorded from the empty region 0.5 μm away without cell presence; which is a remarkable photothermal infrared response providing for direct evidence of the submicron spatial resolution capability with O-PTIR spectroscopy.

Comparison of results collected for 20 infected and 20 control RBCs (FTIR microspectroscopy vs O-PTIR)

as it is difficult to conclude whether lack/or presence of visible differences among spectra collected only for the same single cell (control RBC) can be treated as a general trend for all analyzed cells, wider perspective is needed to draw any truthful conclusion. Analysis

performed on: 124 spectra collected from uninfected (control) and 97 spectra from infected RBCs (iRBC - the trophozoite phase) in case of FTIR microspectroscopy, for O-PTIR spectroscopy 64 spectra for control and 164 spectra for iRBC were taken into account:

- PCA was done to recognise if there is any clustering pattern among studied spectra.
- comparison of O-PTIR mean spectra collected for control and infected RBCs with the PC1 loading was done in order to find ROI in wavenumbers which could play the role in observed clustering for O-PTIR measurements
- comparison of area calculated under the spectrum line within the region 1725-1700 cm^{-1} for each spectrum collected for control and infected RBCs by means of FTIR microspectroscopy and O-PTIR was done in order to see if the presence of Hz can be found in spectra collected for infected RBCs
- comparison of the relative positions of the amide I bands in single spectra collected for control and infected RBCs by means of FTIR microspectroscopy and O-PTIR was done to check if there is any shift which could point for modification in the secondary structure of proteins composition

AFM experiments (example of results for single infected RBC is presented)

for AFM-IR experiments results - RGB composite image (red channel - topography, green channel -1660 cm^{-1} , blue channel - 1740 cm^{-1}) has been presented to show an advantage of spatial resolution over FTIR and O-PTIR

- single spectra were presented to show quite dynamic biochemistry not only within amide I and II bands but also within the regions assigned to nucleic acids (including the symmetric ($\sim 1080 \text{ cm}^{-1}$) and asymmetric ($\sim 1230 \text{ cm}^{-1}$) phosphodiester vibration) and amide III band ($\sim 1286 \text{ cm}^{-1}$)

-additional PLL (Phase Locked Loop) distribution was presented to show that stiffness within infected RBC may be connected to the area with traces of hemozoin

2) As this manuscript's primary goal is to compare three IR methods, more technique details should be included. For example, it would be beneficial to introduce the basic working principles of these methods and show a schematic illustration of setups.

Response: We thank the Reviewer for this comment. Based on the Reviewers' suggestions, we have re-phrased main objectives of our paper. Furthermore, we have extensively modified and improved the Results section. In the current version of the manuscript, three methods are compared by the results collected for control and infected RBCs.

Working principles of AFM-IR and O-PTIR are discussed in references 24 and 27 added to this manuscript.

24. Dazzi, A. & Prater, C. B. AFM-IR: Technology and applications in nanoscale infrared spectroscopy and chemical imaging. *Chem. Rev.* **117**, 5146–5173 (2017).
27. Kansiz, M. *et al.* Optical Photothermal Infrared Microspectroscopy with Simultaneous Raman – A New Non-Contact Failure Analysis Technique for Identification of $<10 \mu\text{m}$ Organic Contamination in the Hard Drive and other Electronics Industries. *Micros. Today* **28**, 26–36 (2020).

3) More information can be included in Table 1, such as image acquisition speed and laser power required. AFM-IR is supposed to benefit from the field enhancement by the metallic tip. So, is the IR power much smaller than O-PTIR or FTIR?

Response: The conventional thermal (Globar) source of IR light emits with a power of $\sim 80 \mu\text{W}$ over the whole spectral range. While a SR based source does not typically produce more power than a Globar source, its brightness (defined as the photon flux or power emitted per source area and solid angle) is 100–1,000 times greater.

In case of O-PTIR spectroscopy we dealt with the power 1-5 mW per spectrum, for AFM-IR 20% of that power was needed to collect spectrum with good SNR.

We thank the Reviewer for this comment, these additional information was added to the Table 3 Advantages and disadvantages of methods discussed in this manuscript.

4) In Figure 2f, the height color bar is missing. One major advantage of AFM is the capability of extracting height information of samples. From this image, one can see the surface of the cell is relatively rough. Could this be part of the reason for the observed different protein distribution in Figures 5c and 5d? Considering a soft tip was used whose oscillations might be sensitive to the sample's sharp features, it is likely the different distributions come from artifacts. Also, the cell on the right side in Figure 5d shows strong responses on its edge. What could be the reason for this?

Response: Color bar was added to represent the height information, in this revised version of manuscript AFM height channel is presented in Fig.5a.

As was suggested by the Reviewers and Editor, the Results section has been modified and in the current version we do not present Fig. 5c. However in the previous version Fig 5c and Fig.5d show the distribution of the amide I collected by means of O-PTIR (Fig.5c) and AFM-IR (Fig. 5d).

Right side in the Fig. 5d depicts distribution of amide I for another (adjacent) cell.

5) In lines 366 and 367, the authors estimated the area where O-PTIR and AFM-IR spectra were taken. 500 nm and 40 nm are the length or diameter of the area, and they should not be directly used as 500 nm^2 or 40 nm^2 .

Response: We thank the Reviewer for this comment, indeed 500 nm^2 or 40 nm^2 should not be used, as this 500nm and 40nm are rather the diameter of the beam spot. These are corrected in the revised submission.

6) This is trivial, but the authors might want to improve their formatting and figure qualities. Some figures and labels are not aligned.

Response: Thanks for pointing out, we have them fixed in this revised manuscript version.

REVIEWERS' COMMENTS:

Reviewer #2 (Remarks to the Author):

When I wrote the review for the first version of this manuscript, I struck a very harsh tone. While I stand by the content of that review, I do now regret the tone. Especially after having read the revised version. The additional data and discussion make this paper a very valuable contribution to the field.

I am missing info on preprocessing performed on spectra depicted in figure 1. It seems they were smoothed and offset, but I haven't found that information in the text. Once the information on spectral preprocessing has been added, the manuscript can be published as is.

Reviewer #3 (Remarks to the Author):

The authors have addressed all my concerns/questions in their revision. The Results section has been greatly improved. I would now recommend the publication of this manuscript on Commun. Chem.

Reviewer#2

Reviewer #2 (Remarks to the Author):

When I wrote the review for the first version of this manuscript, I struck a very harsh tone. While I stand by the content of that review, I do now regret the tone. Especially after having read the revised version. The additional data and discussion make this paper a very valuable contribution to the field.

I am missing info on preprocessing performed on spectra depicted in figure 1. It seems they were smoothed and offset, but I haven't found that information in the text. Once the information on spectral preprocessing has been added, the manuscript can be published as is.

We thank the Reviewer for all comments/suggestions/doubts. For us, all of them were very constructive. Reading the first review opened our eyes and helped us to improve our manuscript. We are grateful that you saw the potential in our work and did not reject our paper immediately.

Waiting for our paper re-evaluation results was a very stressful moment for us as the submitted version differed greatly from the previous one. Seeing your comments (especially this expression)

“The additional data and discussion make this paper a very valuable contribution to the field” was very rewarding to us. Thank you so much for your support, for giving us a helpful hand.

As for spectra depicted in Fig. 1, they were only offset, no smoothing was applied in pre-processing. Fig. 1 caption was modified accordingly.

Reviewer #3 (Remarks to the Author):

The authors have addressed all my concerns/questions in their revision. The Results section has been greatly improved. I would now recommend the publication of this manuscript on Commun. Chem.

Thank you so much for all your comments and suggestions; they helped us to improve quality of our paper. We do appreciate your time in re-evaluation of our paper and your recommendation for publishing our work in Communications Chemistry.